# Sexual Competitiveness and Induced Egg Sterility by *Aedes aegypti* and *Aedes albopictus* Gamma-Irradiated Males: A Laboratory and Field Study in Mexico

**DOI:** 10.3390/insects12020145

**Published:** 2021-02-08

**Authors:** J. Guillermo Bond, Santiago Aguirre-Ibáñez, Adriana R. Osorio, Carlos F. Marina, Yeudiel Gómez-Simuta, Rodolfo Tamayo-Escobar, Ariane Dor, Pablo Liedo, Danilo O. Carvalho, Trevor Williams

**Affiliations:** 1Centro Regional de Investigación en Salud Pública (CRISP-INSP), Tapachula, Chiapas 30700, Mexico; cruzdsantiago@hotmail.com (S.A.-I.); dris97@hotmail.com (A.R.O.); fmarina@insp.mx (C.F.M.); 2Programa Moscas de la Fruta (SADER-IICA), Metapa de Domínguez, Chiapas 30860, Mexico; yeudiel.simuta.i@senasica.gob.mx (Y.G.-S.); rodolfo.tamayo.i@senasica.gob.mx (R.T.-E.); 3Consejo Nacional de Ciencia y Tecnología (Cátedras) commissioned to El Colegio de la Frontera Sur, Tapachula, Chiapas 30700, Mexico; ador@ecosur.mx; 4El Colegio de la Frontera Sur (ECOSUR), Tapachula, Chiapas 30700, Mexico; pliedo@ecosur.mx; 5Insect Pest Control Laboratory, Joint FAO/IAEA Programme of Nuclear Techniques in Food and Agriculture, IAEA Laboratories, 2444 Seibersdorf, Austria; d.carvalho@iaea.org; 6Instituto de Ecología AC (INECOL), Xalapa, Veracruz 91073, Mexico; trevor.williams@inecol.mx

**Keywords:** sterile insect technique, SIT, mosquito vector, arboviruses, egg viability, egg production, insemination

## Abstract

**Simple Summary:**

The sterile insect technique (SIT) involves the release of massive numbers of male insects that have been sterilized by irradiation treatment during their development. Wild females that mate with sterilized males are not able to produce offspring, resulting in rapid decline in the target insect population over a large area. The success of this technique depends on the ratio of wild:sterile males achieved following male releases and the ability of sterile males to mate with wild females, i.e., their sexual competitiveness compared to fertile wild male insects. There is growing interest in applying SIT to the area-wide control of mosquitoes, such as *Aedes aegypti* and *Aedes albopictus*, that transmit important human diseases caused by dengue, chikungunya, and Zika viruses. In the present study, the sexual competitiveness of both mosquito species was affected by irradiation treatments but did not vary greatly with different ratios of fertile:sterile males in mating cages. Most importantly, the fertility of eggs was greatly reduced when more sterile males were present in mating cages, resulting in an 88% decrease in the production of fertile eggs by both species of mosquitoes in some experiments. We will use these results to perform small-scale trials in rural villages frequently affected by outbreaks of mosquito-borne diseases in southern Mexico.

**Abstract:**

The sterile insect technique may prove useful for the suppression of mosquito vectors of medical importance in regions where arboviruses pose a serious public health threat. In the present study, we examined the effects of sterilizing irradiation doses across different ratios of fertile:irradiated males on the mating competitiveness of *Ae. aegypti* and *Ae. albopictus* under laboratory and field-cage conditions. For both species, the percentage of females inseminated and the number of eggs laid over two gonotrophic cycles varied significantly in mating treatments involving 1:1, 1:5, and 1:10 fertile:irradiated males compared to controls of entirely fertile or entirely irradiated males but was not generally affected by the irradiation dose. Egg hatching was negatively affected in females exposed to increasing proportions of irradiated males in both laboratory and field cages. Male competitiveness (Fried’s index) values varied from 0.19 to 0.58 in the laboratory and were between 0.09 and 1.0 in field cages, depending on th species. Competitiveness values were negatively affected by th eirradiation dose in both species under field-cage conditions, whereas in the laboratory, *Ae. albopictus* was sensitive to the dose but *Ae. aegypti* was not. In general, male competitiveness was similar across all mating regimes. Most importantly, induced egg sterility was positively correlated with the proportion of irradiated males present in the mating treatments, reaching a maximum of 88% under field-cage conditions for both *Ae. aegypti* and *Ae. albopictus* males treated with 50 and 40 Gy irradiation, respectively. These results indicate that sterile males produced at our facility are suitable and competitive enough for field pilot SIT projects and provide guidance to decide the optimal sterile:fertile ratios.

## 1. Introduction

Although numerous viruses are transmitted by mosquitoes, the dengue, yellow fever, chikungunya, and Zika viruses have caused the most human suffering in tropical and sub-tropical regions of the Americas [1]. *Aedes aegypti* is the primary vector of these arboviruses, whereas the invasive Asian tiger mosquito, *Ae. albopictus*, is of secondary importance [2,3]. In the absence of effective vaccines, interruption of the transmission cycles of these viruses mainly involves control of vector populations in urban areas [4]. Control of *Aedes* vectors has mainly involved the elimination of containers that are favorable sites for oviposition and development of mosquito aquatic stages (source reduction) and chemical control measures involving larvicidal treatment of water sources and the use of adulticides in outdoor and indoor residential settings and in recreational, commercial, and industrial areas. Frequent exposure to insecticides is often associated with adverse effects on non-target organisms and the evolution and spread of resistance to these compounds, which is a major concern for vector control programs [5].

In Mexico, over 41,000 confirmed cases and more than 268,000 suspected cases of dengue were officially recorded in 2019, the majority of which occurred in the states of Jalisco, Veracruz, and Chiapas [6]. These figures are likely to be an underestimate as many people with dengue do not seek medical attention until the symptoms of the disease become severe. The Zika and chikungunya viruses are also endemic in Mexico since their arrival in 2014 [7]. Given this situation, the need for new, effective, and sustainable vector management tools in this region has become increasingly evident.

Area-wide programs based on the sterile insect technique (SIT) have demonstrated several successes in the control of agricultural pests, and there are good prospects for their successful application in the public health field [8,9,10,11,12]. SIT is a species-specific and environmentally benign pest control strategy that relies on mass rearing, sterilization, and area-wide release of large numbers of sterile males that outcompete wild males for mates [13,14]. The radiation-induced chromosomal damage to sperm means that wild females that mate with irradiated males produce infertile eggs, resulting in pest population reduction and elimination in some cases [13,15]. In mass production facilities, males are subjected to unnatural conditions that usually include a semi-synthetic diet, constant controlled environmental conditions, high rearing densities, reduced genetic diversity of the insect colony, and manipulation during sterilization. These factors can affect the overall quality of the male insects produced, their field survival, and their ability to compete with wild males for mating with wild females [13,16,17].

The effects of irradiation dosage on insect fertility, longevity, and sexual competitiveness need to be clearly understood as they each have a direct influence on the efficacy of SIT-based control programs [18,19]. For example, the quality of irradiated males can be adversely affected by radiation-induced damage to somatic cells [12], resulting in reduced survival and sexual competitiveness of sterile males when compared to wild males [20,21]. In a previous study, we determined the irradiation dose–fertility response in Mexican colonies of *Ae. aegypti* and *Ae. albopictus* when irradiated in the pupal stage [18]. Doses of 50 and 70 Gy resulted in 0.6% and 0% male fertility, respectively, in *Ae. aegypti*, whereas doses of 40 and 60 Gy resulted in 0.9% and 0% fertility, respectively, in *Ae. albopictus* males. Adult mosquito survival and flight ability were not adversely affected by these doses of irradiation under laboratory conditions in either species [18].

Having established the range of sterilizing doses of irradiation for these species [18], the present study aimed to assess the effects of sterilizing irradiation doses on mating competitiveness and induced egg sterility in *Ae. aegypti* and *Ae. albopictus* at differing ratios of fertile males:irradiated males in both laboratory and field-cage conditions. The study was performed on the Mexican populations of these mosquitoes as irradiated male mating competitiveness can differ between insect populations, so the findings from one region may differ quantitatively or qualitatively from those in geographically distant regions [11,22]. As such, the results of the present study will serve to estimate the minimum required release ratio of sterile males to wild males that could suppress the natural populations of these vectors in a dengue-endemic region of southern Mexico.

## 2. Materials and Methods

### 2.1. Mosquito Strains

The *Aedes aegypti* strain used in the experiments was a genetically diverse strain (GDS1) collected as eggs at 12 sites along the Pacific coast of Chiapas State, Mexico. The genetically diverse *Ae. albopictus* strain (GDS2) was also collected from four sites along the Pacific coast of Chiapas State [18].

Larvae were reared at a density of 1.5 insects/mL in 61 × 41 × 7.5 cm^3^ plastic trays containing 2000 mL of dechlorinated water and were fed with liquid Laboratory Rodent Diet (LabDiet, Fort Worth, TX, USA), as described previously [23]. Pupae were separated by sex as a function of body size using a plate separator (John W. Hock, Model 5412, Gainesville, FL, USA) and confirmed by examination of the genital lobe using a Stemi 508 stereomicroscope (Carl Zeiss, Oberkochen, Germany). Both colonies were maintained under controlled conditions at 28 ± 2 °C for larvae and 26 ± 2 °C with 80 ± 5% relative humidity (RH) for adults, and a photoperiod of 14:10 h (light:darkness (L:D)) for both stages.

### 2.2. Pupae Irradiation

The irradiator used was a dry storage irradiator (Gamma Beam GB-127, serial number IR-226; Nordion, Ottawa, ON, Canada), with a cobalt-60 (^60^Co, activity 14416 Ci) source located in the Moscafrut facility in Metapa, Chiapas, Mexico. The dose rate was determined using an ionization chamber RADCAL™ Model ADDM (RADCAL, Monrovia, CA, USA). Dosages were determined using the Fricke dosimetry system [24] and a Gafchromic film dosimetry system [25]. Male pupae were irradiated 24–36 h before adult emergence, as described previously [18]. Calibrated doses of 50 and 70 Gy were obtained by placing *Ae. aegypti* pupae at distances of 54 and 43 cm from the source, respectively, over a 10 min period. Irradiation of *Ae. albopictus* was performed when the ^60^Co source had been recharged and recalibrated, and doses of 40 and 60 Gy were obtained by placing pupae at distances of 75 and 59 cm, respectively, over a 5 min period. For each species and each dose, batches of 2000 pupae were placed in 50 mL of dechlorinated water in a plastic tray 10 cm in diameter and 4 cm high. Three independent batches of insects were irradiated for each treatment.

### 2.3. Competitiveness Experiments

#### 2.3.1. Laboratory Cages

Laboratory cage observations on both species were performed at 26 ± 2 °C with 80 ± 5% relative humidity (RH) and a 14:10 h L:D photoperiod. Cages of 30 × 30 × 30 cm^3^ comprised an acrylic frame with nylon mesh walls (BugDorm 1, Taichung, Taiwan). To examine male sexual competitiveness in laboratory cages, insects were randomly assigned to one of five treatments: Hn (fertile male controls) in which 50 fertile females with 50 fertile males (1:1 ratio) were exposed; Hs (sterile male controls) in which 50 fertile females with 50 sterile males (1:1 ratio) were introduced; Ho1, in which 50 fertile females, 50 fertile males, and 50 sterile males (1:1:1 ratio) were introduced; Ho5, which consisted of 50 fertile females, 50 fertile males, and 250 sterile males (1:1:5 ratio); and finally Ho10, which consisted of 50 fertile females, 50 fertile males, and 500 sterile males (1:1:10 ratio). All insects were 5 days old at the start of the experiment. In all cases, the cage size remained constant despite changes in mosquito densities in the different mating treatments.

In all treatments, males were gently released into the cage, followed 1 h later by the release of females. Insects were allowed to mate for 24 h following female introduction. Following this, a group of 10 females from each irradiated batch were selected at random and checked for the presence of spermatozoa in the spermathecae under a microscope at 400 × magnification (Primo Star, Carl Zeiss, Jena, Germany). In this way, the percentage of inseminated females was estimated for each batch of insects. The remaining females were then transferred to another cage (30 × 30 × 30 cm^3^) and fed with bovine blood for 30 min using a Hemotek membrane feeding system (PS6B, Hemotek Ltd., Great Harwood, UK). At 3 days after feeding, an oviposition container (500 mL capacity, 11-cm-diameter × 9-cm-height plastic containers with 200 mL of water and a 32 × 5 cm filter paper strip for oviposition) was placed in each cage for a 48 h period, after which paper strips with eggs were removed and allowed to dry. Females were allowed to feed again at 5 days after the first blood meal, and eggs were collected in oviposition containers as before. The number of eggs on paper strips was counted and recorded, and eggs were hatched in a 500 mL glass jar with an airtight lid, which was filled with water that had been boiled and allowed to cool, into which the eggs on paper strips were placed for 2 h. After 2 days, the number of hatched and unhatched eggs and the number of larvae were counted. In this way, egg production and the percentage of egg hatching (an indicator of fertility) were determined over two gonotrophic cycles.

#### 2.3.2. Field-Cage Test

The field-cage study was performed in two different periods: from July to November 2019 for *Ae. aegypti* and from April to August 2020 for *Ae. albopictus*. Field cages consisted of a 2 × 2 × 2 m^3^ cube of nylon mosquito netting suspended from steel tubes. These cages were placed in a large field cage 30 × 8.5 × 6.5 m^3^ (length × width × height) constructed from steel tubes and covered with anti-insect mesh (10 threads/cm) with a 70% shade cloth hung inside the roof to reduce incident sunlight. The cage was located in a pasture (14° 50′38.22″ N, 92°20′11.95″ W) near the village of Río Florido, Chiapas, southern Mexico, and was surrounded by palms and mango orchards.

Climatic conditions were measured using a data logger (Hobo U12–013, Onset, Bourne, MA, USA). The study was performed at an average temperature of 29.8 ± 4 °C (minimum and maximum range 23.2–38.7 °C) and 77.5 ± 13% RH (range 51.3–94.0%) for *Ae. aegypti* and 30.1 ± 4 °C (range 23.4–37.5 °C) and 76.2 ± 13% RH (range 50.6–93.4%) for *Ae. albopictus*. The photoperiod was approximately 12:12 h L:D for both species.

The experiment in field cages involved similar treatments as described in the laboratory cage study, except for the numbers of insects present in the cages. For the calculation of competitiveness, the following treatments were applied: Hn (fertile male controls), in which 100 fertile females with 100 fertile males (1:1 ratio) were exposed; Hs (irradiated male controls), in which 100 fertile females with 100 irradiated males (1:1 ratio) were introduced; Ho1, in which 100 fertile females, 100 fertile males, and 100 irradiated males (1:1:1 ratio) were introduced; Ho5, which consisted of 100 fertile females, 100 fertile males, and 500 irradiated males (1:1:5 ratio); and finally Ho10, which consisted of 100 fertile females, 100 fertile males, and 1000 irradiated males (1:1:10 ratio). All insects were 5 days old at the beginning of the experiment. Three repetitions (cages) were performed for each treatment and for each species. In all cases, females were released into cages in which males had been released 1 h previously. Mosquitoes had continuous access to 10% sucrose solution in a 250 mL container with a cotton wick placed on the ground in the center of each cage. After the 24 h mating period, females were carefully collected using an aspirator, taken to the laboratory in an insulated box, counted, and placed in acrylic cages of 30 × 30 × 30 cm^3^ with nylon mesh walls (BugDorm 1; Taichung, Taiwan), with access to 10% sucrose solution on a cotton pad. To determine the percentage of inseminated females, a randomly selected group of 10 females from each treatment was dissected and checked for the presence of spermatozoa. The remaining females were offered a blood meal 24 h after collection from field cages, and eggs were collected and counted over two gonotrophic cycles, as described in the laboratory-cage experiment. The average number of eggs per female was obtained by dividing the total number of eggs by the number of females present in each cage. Similarly, collected eggs were dried and then allowed to hatch to determine the percentage of egg hatching, as described in the laboratory test.

The male mating competitiveness index (*C*) was calculated using the Fried (1971) equation [26]:*C* = (*Hn* − *Ho*)/(*Ho* − *Hs*) ∗ (*N/S*)
where *Hn* is the percentage of egg hatch from eggs of females that mated with fertile males; *Hs* is the percentage of egg hatch from eggs of females that mated with irradiated males; *Ho* is the observed percentage of egg hatch in each of the mating treatments (involving 1:1, 1:5, and 1:10 fertile:irradiated males); *N* is the number of fertile males released at the start of the experiment; and *S* is the initial number of irradiated males. The percentage of induced sterility (*IS*) was calculated using the equation:*IS* = (1 − [*Ho*/*Hn*]) ∗ 100

### 2.4. Statistical Analysis

The percentage of egg hatch, percentage of female insemination, total egg production (fecundity), competitiveness index, values and percentage of induced sterility values recorded in laboratory and field-cage conditions were subjected to two-way analysis of variance (ANOVA) with mating treatment and irradiation dose as fixed factors. Interaction terms were not reported unless they were significant (*p* ≤ 0.05). Means of egg hatch, insemination, and egg production were compared by the Bonferroni test, whereas mean competitiveness and induced sterility values were compared by the Tukey test. Data were checked for normality and homoscedasticity prior to analysis by applying Shapiro–Wilk and Levene’s tests, respectively. Where necessary, variance was controlled by rank transformation prior to analysis. Each species was analyzed separately using an R-based package [27].

## 3. Results

### 3.1. Laboratory Study

#### 3.1.1. Insemination of Females

The prevalence of insemination of *Ae. aegypti* females, revealed by dissection, varied significantly with mating treatment (*F*_4,20_ = 3.040, *p* = 0.041) and according to dose (*F*_1,20_ = 5.143, *p* = 0.035) (Table 1). In all cases, the prevalence of insemination of females in the fertile male controls (Hn) (83% at both 50 and 70 Gy) was significantly higher than that observed in the irradiated male controls (Hs) (50–63%, depending on dose), with intermediate values in the treatments involving different proportions of fertile and irradiated males (Table 1).

The insemination of *Ae. albopictus* females under laboratory conditions differed significantly among treatments (*F*_4,20_ = 11.18, *p* < 0.0001) but not between doses (*F*_1,20_ = 1.64, *p* = 0.215) (Table 1). The percentage of inseminated females in the irradiated male controls (Hs) varied between 43% and 50%, depending on dose, and was significantly lower than that observed in the fertile male controls (Hn, 73–77%). Insemination of females in the other treatments was similar to that of the fertile controls except for the Ho1 treatment (1:1 fertile:irradiated males) with 53–57% of females inseminated, which was similar to that of the irradiated male controls (Hs) at both doses (Table 1). These results indicate that irradiation tends to reduce the mating ability of both *Ae. aegypti* and *Ae. albopictus* compared to untreated fertile males under laboratory conditions.

#### 3.1.2. Egg Production

Of the *Ae. aegypti* females released into mating cages, overall 94.6 ± 0.99% (range 70–100%) of females were alive after the 24 h mating period and were used for the egg production study. The mean number of eggs produced by *Ae. aegypti* in laboratory cages, averaged over two gonotrophic cycles, differed significantly among mating treatments (*F*_4,20_ = 6.071, *p* = 0.0023) but not between doses (*F*_1,20_ = 0.238, *p* = 0.6308) (Table 1A). Egg production across all treatments was similar to that of fertile control females (mean 34.4–34.8 eggs/female, depending on dose), whereas egg production was lowest in females that mated with irradiated male controls (mean 24.0–31.3 eggs/female, depending on dose).

Of the *Ae. albopictus* females released into cages, overall 91.7 ± 1.42% (range 40–100%) of females were recovered alive after mating and were used for the egg production study. Egg production by *Ae. albopictus* in laboratory cages differed significantly among treatments (*F*_4,20_ = 2.981, *p* = 0.0441), whereas the dose was not significant (*F*_1,20_ = 1.904, *p* = 0.1828) (Table 1B). The production of eggs was lowest in females that mated with irradiated males (Hs treatment, 20.0–23.9 eggs/female, depending on dose) and highest in the fertile male controls (Hn, 34.2–44.4 eggs/female) and Ho10 treatment (1:10 fertile:irradiated males; 41.3–42.8 eggs/female), with intermediate values in the other treatments (Table 1B). These results indicate that females exposed to irradiated males generally produce fewer eggs than females exposed to fertile males, possibly due to a reduced prevalence of insemination observed in the previous section.

#### 3.1.3. Egg Hatch

The mean egg hatch of *Ae. aegypti*, an indicator of egg fertility, differed significantly among treatments of irradiated mosquitoes and the controls (*F*_4,20_ = 102.61, *p* < 0.0001) but not between doses (*F*_1,20_ = 0.43, *p* = 0.517) or the dose–treatment interaction (ANOVA, *F*_4,20_ = 0.10, *p* = 0.983) (Figure 1A). Egg hatch in the fertile male controls (Hn) varied between 83.7% and 87.6%, depending on dose, but steadily decreased in the other treatments as the fraction of irradiated males increased to a minimum of less than 0.1% in the irradiated male controls (Hs) (Figure 1A).

In *Ae. albopictus*, the mean egg hatch rate differed significantly across the treatments (*F*_4,20_ = 186.19, *p* < 0.0001) but not between doses (*F*_1,20_ = 0.86, *p* = 0.3656) (Figure 1B). Egg hatch in the fertile male controls (Hn) was 79.7–82.8%, depending on dose, but was consistently lower in the treatments involving increasing numbers of irradiated males, and it was lowest in the irradiated male controls (Hs, <0.2% in both doses).

#### 3.1.4. Male Competitiveness and Induced Egg Sterility

The competitiveness index values of *Ae. aegypti* males in laboratory cages were rank-transformed to control variances prior to analysis. Competitiveness values varied between 0.24 and 0.55 but were not significantly affected by irradiation dose (*F*_1,12_ = 0.038, *p* = 0.849) or mating treatment (*F*_2,12_ = 0.607, *p* = 0.561) (Figure 2A), indicating that *Ae. aegypti* males were not adversely affected by irradiation under laboratory mating conditions. The percentage of induced sterility of eggs was not significantly affected by irradiation dose (*F*_1,12_ = 0.029, *p* = 0.867) but increased significantly from approximately 18% sterility in the Ho1 mating treatment (1:1 fertile:irradiated males) to 76–79% in the Ho10 treatment (1:10 fertile:irradiated males), with intermediate values (~52%) in the Ho5 treatment (*F*_2,12_ = 31.4, *p* < 0.001) (Figure 2A), indicating that egg sterility was directly correlated with the ratio of fertile:irradiated males in laboratory cages.

The competitiveness of *Ae. albopictus* males in laboratory cages was significantly reduced in males that received the higher dose of radiation (60 Gy, index values in the range 0.19–0.40) compared to those that received the 40 Gy dose (index values of 0.44–0.58) (*F*_1,12_ = 6.912, *p* = 0.022) (Figure 2B). Competitiveness was not significantly affected by mating treatment (*F*_2,12_ = 1.100, *p* = 0.364). The percentage of induced egg sterility was significantly higher in cages containing males that received the 40 Gy dose compared to those that received the 60 Gy dose (*F*_1,12_ = 10.123, *p* = 0.008) (Figure 2B). Egg sterility values increased significantly with an increasing ratio of fertile:irradiated males from 27.6–34.1% in the Ho1 treatment to 63.3–78.8% in the Ho10 treatment (*F*_2,12_ = 29.274, *p* < 0.001).

### 3.2. Field-Cage Conditions

#### 3.2.1. Insemination of Females

The prevalence of insemination of *Ae. aegypti* females in field cages varied significantly with mating treatment (*F*_4,20_ = 3.330, *p* = 0.0303) but not with dose (*F*_1,20_ = 2.717, *p* = 0.1149) (Table 2A). Similar to the results of laboratory cages, the prevalence of insemination of females fell from 73–83%, depending on dose, in the fertile male controls (Hn) to 53–60% in the irradiated male controls (Hs), with statistically similar values in the different mating treatments (Table 2A).

The insemination of *Ae. albopictus* females under field-cage conditions was also significantly affected by the mating treatment (*F*_4,20_ = 6.34, *p* < 0.0018) but not by dose (*F*_1,20_ = 0.29, *p* = 0.5956) (Table 2B). Insemination in fertile females (Hn) was 60–70%, depending on dose, but was significantly lower in females that mated only with irradiated males (Hs) and averaged 40–37%, depending on dose, with intermediate values in the other mating treatments and slightly higher values in the Ho10 treatment (Table 2B).

#### 3.2.2. Egg Production

For *Ae. aegypti*, the recovery of living females after mating was 98.0 ± 0.29% (range 94–100%), and these females were taken to the laboratory, given a blood meal, and used for egg production studies. The mean egg production of *Ae. aegypti* that had mated in field cages differed significantly among mating treatments (*F*_4,20_ = 5.254, *p* = 0.0047) but did not differ significantly with dose (*F*_1,20_ = 2.956, *p* = 0.1010) (Table 2A). The mean number of eggs produced by females in the fertile male controls (Hn), averaged over two gonotrophic cycles, was 19.2–21.9 eggs/female and was similar in all the mating treatments, except for the irradiated male controls (Hs), in which mean egg production was significantly reduced at 11.6–15.2 eggs/female (Table 2A).

Recovery of *Ae. albopictus* females from field cages were 77 ± 2% (range 41–96%), and these females were used for the egg production study. Egg production by *Ae. albopictus* females that had mated in field cages differed significantly among treatments (*F*_4,20_ = 14.640, *p* < 0.0001) and was also significantly affected by irradiation dose (*F*_1,20_ = 9.898, *p* = 0.0051). Egg production was lower in females that mated with males irradiated at 60 Gy compared to males irradiated at 40 Gy (Table 2B). For both doses, egg production was highest in the fertile male controls (Hn; 30.4 and 24.1 eggs/female in the 40 and 60 Gy controls, respectively) and lowest in the irradiated male controls (Hs; 15.2 and 8.5 eggs/female at 40 and 60 Gy treatments, respectively), with intermediate values in the remaining mating treatments (Table 2B).

#### 3.2.3. Egg Hatch

Egg hatch in *Ae. aegypti* that mated under field-cage conditions varied significantly with mating treatment (*F*_4,20_ = 87.57; *p* < 0.0001) but not with dose (*F*_1,20_ = 2.37, *p* = 0.1392), although the treatment–dose interaction was significant (*F*_4,20_ = 2.92, *p* = 0.0473) (Figure 3A). The percentage of egg hatch steadily decreased from 79.6–86.7% in the fertile male controls (Hn) to 0.3–0.4% in the irradiated male controls (Hs), with decreasing values in the other treatments (Figure 3A). The interaction effect appeared to involve the low egg hatch value (28.2%) observed in the Ho5 (1:5 fertile:irradiated males), 50 Gy treatment, compared to that in the Ho5, 70 Gy treatment (51.7%).

In the case of *Ae. albopictus*, egg hatch was significantly affected by the mating regime (*F*_4,20_ = 173.11, *p* < 0.0001), dose (*F*_1,20_ = 30.73, *p* < 0.0001), and the treatment–dose interaction (*F*_4,20_ = 6.81, *p* = 0.0013) (Figure 3B). The overall decreasing trend in the percentage of egg hatch values seen in the laboratory study was maintained in the field-cage study, with the highest values in the fertile male controls (Hn) (85.1–90.7%, depending on dose) and the lowest values in the irradiated male controls (Hs) (0.3–0.4%, depending on dose). Egg hatch was significantly lower at the 40 Gy dose in treatments involving Ho5 and Ho10 (1:5 and 1:10 fertile:irradiated males, respectively) compared to the same treatments that were irradiated with 60 Gy (Figure 3B).

#### 3.2.4. Male Competitiveness and Induced Egg Sterility

A single replicate in the 70 Gy Ho1 treatment of *Ae. aegypti* males that mated in field cages produced negative competitiveness and induced sterility values and was removed from the analysis. The competitiveness index values of *Ae. aegypti* males were rank-transformed to control variance prior to analysis. Mean competitiveness values of males that received the 50 Gy dose (range 0.41–1.0) were significantly higher than those that received the 70 Gy dose (range 0.09–0.46) (*F*_1,11_ = 7.870, *p* = 0.017) but did not vary significantly among mating treatments (*F*_2,11_ = 1.460, *p* = 0.274) (Figure 4A). The percentage of induced sterility was significantly higher at the 50 Gy dose compared to the 70 Gy dose (*F*_1,11_ = 12.368, *p* = 0.005). Egg sterility also increased significantly as the proportion of irradiated males increased (*F*_2,11_ = 19.313, *p* < 0.001) from a minimum of 15.5–29.9% in the Ho1 treatment to 64.4–88.0% in the Ho10 treatment (Figure 4A).

The competitiveness of *Ae. albopictus* males in field cages was significantly reduced in males that received the higher dose of radiation (60 Gy, index values in the range 0.12–0.25) compared to those that received the 40 Gy dose (index values 0.31–0.80) (*F*_1,12_ = 18.635, *p* = 0.001) (Figure 4B). Competitiveness was significantly affected by mating treatment (*F*_2,12_ = 3.962, *p* = 0.048), and the treatment–dose interaction (*F*_2,12_ = 6.733, *p* = 0.011) was indicative of a significantly higher competitiveness value in the 40 Gy Ho10 treatment (*C* = 0.80) compared to the lower values observed in the other treatment combinations (Figure 4B). The percentage of induced egg sterility was significantly higher in field cages containing *Ae. albopictus* males that received the 40 Gy dose compared to those that received the 60 Gy dose (*F*_1,12_ = 17.166, *p* = 0.001) (Figure 4B). Egg sterility values also increased significantly with the increasing proportion of irradiated males from 18.9–22.4% in the Ho1 treatment to 52.2–88.3% in the Ho10 treatment (*F*_2,12_ = 32.775, *p* < 0.001).

## 4. Discussion

Following several decades of successes in the area-wide control of insect pests of agricultural and veterinary importance, the potential applications of SIT-based programs are now attracting growing attention for the control of insect vectors of human arboviruses [9,11]. In the present study, we examined the effects of two irradiation doses on the sexual competitiveness of Mexican populations of *Ae. aegypti* and *Ae. albopictus* males in both laboratory and field-cage conditions. The induced egg sterility was also examined across three ratios of wild to irradiated males to provide an indicator of the release ratio required to control natural populations of these vectors in southern Mexico.

Females insemination tended to be highest in the fertile male controls (Hn) and lowest in the irradiated male controls (Hs) in both laboratory and field-cage conditions (Table 1 and Table 2), usually with intermediate values in the different mating treatments (Ho1–Ho10). The prevalence of female insemination also tended to be slightly higher in *Ae. aegypti* than in *Ae. albopictus*. In laboratory cages, irradiation reduced the insemination capacity of males of *Ae. aegypti*, suggesting an overall impact on male vigor or attractiveness, an effect that was not observed in *Ae. albopictus* males. However, no dose effects on female insemination were detected in field cages; here insemination was usually lower than that observed in the laboratory, especially in the case of *Ae. albopictus*. In all cases, the prevalence of insemination increased consistently with increasing numbers of irradiated males in the mating treatments (Ho1–Ho10). Similar results were observed in the prevalence of female insemination in the fertile controls of *Anopheles arabiensis* and *An. coluzzii* exceeded that of the irradiated control and increased with increasing prevalence of irradiated males (1:1, 1:5 and 1:10) [21,28]. In a longitudinal study, the insemination capacity of irradiated *Ae. albopictus* males was strongly age dependent, with a clearly reduced capacity early (1 day old) and late (9 days old) in adult life [29], although this was not investigated in the present study in which insect age was controlled in all experiments.

In general, egg production reflected the prevalence of female insemination in both laboratory and field cages (Table 1 and Table 2). Females of *Ae. aegypti* and *Ae. albopictus* that mated with irradiated male controls (Hs) produced fewer eggs than females that mated with fertile male controls (Hn), with intermediate and rather variable egg production values in the different mating treatments (Ho1–Ho10). The dose of irradiation had no significant effect on egg production values except in the case of *Ae. albopictus* that mated in field cages, in which egg production was lower in females that mated with males irradiated at the higher dose (60 Gy), as observed previously in this species [30] and in *An. arabiensis* [31]. A significant decrease in egg production was also observed in a study on Mexican and Brazilian strains of *Ae. aegypti*, in which males were irradiated with 40–50 Gy [32]. The principal cause of reduced egg production by females mated with irradiated males is likely to be related to the success of insemination (or possibly the quantity of semen-related factors transferred during copulation), which could affect female fecundity or the female’s blood-feeding behavior [33,34].

The viability (hatching) of eggs was 80–90% in the fertile male controls (Hn) and decreased steadily with the increasing ratio of fertile to irradiated males to a minimum of less than 1% viability in the irradiated male controls for both species in laboratory and field cages (Figure 1A,B and Figure 3A,B). This trend was not affected by irradiation dose, except for the field-cage results for *Ae. albopictus*, in which egg hatching was significantly reduced in the 40 Gy treatment. These results are consistent with those reported for *Ae. albopictus* in Italy [35] and China [36] or by us in a previous study on our colonies of *Ae. albopictus* and *Ae. aegypti*, in which egg viability was reduced to <1% at doses of 40 and 50 Gy, respectively [17]. Similar, albeit less dramatic, results were also observed in strains of *An. coluzzii* and *An. arabiensis* originating from Burkina Faso and North Sudan, respectively [21,34]. However, the reason why the lower irradiation dose (40 Gy) resulted in lower egg hatch than the higher (60 Gy) dose in *Ae. albopictus* in the present study is unclear, as the percentage of female insemination was not dose sensitive in our field-cage experiment. Indeed, reductions in male biological quality are usually related to the mass-rearing, handling, and insect release processes rather than radiation treatment per se [37].

Male competitiveness varied from 0.19 to 0.58 in the laboratory and between 0.09 and 1.0 in field cages, depending on the species. Competitiveness values were negatively affected by the irradiation dose in both species under field-cage conditions, whereas in the laboratory, competitiveness of *Ae. albopictus* males was sensitive to dose but this was not the case for *Ae. aegypti*. In general, male competitiveness was not affected by the mating regime except for *Ae. albopictus* in field cages, in which competitiveness varied among the mating treatments but without a consistent trend. A negative relationship between male competitiveness and irradiation dose has been observed in several times in studies on mosquitoes, including *Aedes* spp. [12,21,32,36,38,39,40]; however, irradiated males have been shown to be competitive after release in several studies [37].

Reductions in competitiveness can be compensated by adjusting the dose and timing of irradiation treatment or by increasing the proportion of irradiated males released with respect to the fertile male population [12,17,36,37,41,42]. The colonization process, mass-rearing conditions, and irradiation can all negatively impact the competitiveness of males released in a SIT program [11,21,28,32,34,37]. The developmental stage at which males are irradiated can also affect competitiveness as pupae are more susceptible to somatic damage than adults [41,43]. In the present study, high competitiveness values were observed in the field cages and at the ratio of 1:10 fertile:irradiated males for which values varied between 1.0 for *Ae. aegypti* (50 Gy) and 0.8 for *Ae. albopictus* (40 Gy). Previous studies on *Ae. albopictus* have estimated competitiveness values in the range of 0.4–1.0 following irradiation doses of 28–40 Gy; in all cases, competitiveness was higher at lower irradiation doses [12,29,36,42]. For *Ae. aegypti*, competitiveness ranged between 0.70 and 1.37 following treatment of a *Wolbachia*-infected strain of the mosquito with a dose of 70 Gy [8].

In a multi-year study in Italy, tests were performed in field cages containing a constant density of 100:100:100 females: fertile males:irradiated males of *Ae. albopictus* [19]. Irradiated males had been partially sterilized with doses of 30 and 40 Gy, resulting in high competitiveness index values of 1.0–0.72 (Fried’s index) and induced egg sterility values of 0.96–0.71, respectively. In contrast, earlier studies performed using males treated with doses of 50–60 Gy were adversely affected by early adult male emergence that hindered accurate assessment of male competitiveness. This led the authors to conclude that although the lower doses of irradiation resulted in partial male sterility, this was more than compensated for by the high competitiveness and capacity to induce egg sterility in males treated with doses of 30–40 Gy [12]. This finding clearly resonates with our field-cage observations that males of both *Aedes* species had an increased capacity to induce egg sterility when treated with the lower dose of irradiation (Figure 4A,B).

Induced sterility is the prevalence of egg sterility resulting from females that undergo mating in the presence of irradiated males [28,31]. As such, it is the key driver of population decline in areas in which irradiated males have been released. Induced sterility was lower in treatments involving higher doses of irradiation in laboratory and field cages in both species, except for *Ae. aegypti* that mated in the laboratory for which the dose had no significant effect (Figure 2A). In all cases, the prevalence of induced sterility increased with the ratio of fertile:irradiated males in the different mating treatments. Our field-cage studies indicated that 88% of induced egg sterility was achieved in the 1:10 fertile:irradiated male treatment (Ho10) in both *Ae. aegypti* and *Ae. albopictus* at doses of 50 and 40 Gy, respectively, which is a promising result that requires testing in pilot-scale field studies. Others have reported induced sterility of 74% in *Ae. albopictus* at a 1:7 fertile:irradiated male mating regime [36] or up to 79–88% in Mexican and Brazilian strains of *Ae. aegypti* that had been infected with *Wolbachia* prior to irradiation [32].

The irradiation process that is the standard method for producing sterile male insects for release in SIT programs has to be managed effectively if high-quality, sexually competitive males are to be produced in massive numbers. Potential negative effects are often detectable through reductions in male mating performance, reduced flight capacity (dispersal), and adult longevity in comparison to wild fertile males [9,44]. As reduced irradiation doses invariably result in increased male fertility, the main strategy to address reduced competitiveness in irradiated males is to increase the numbers of irradiated males released in each area, although this implies increased costs for mass production and the frequency of releases [19,21,45]. Our results suggest that by using the lower irradiation doses tested here, increased male competitiveness could outweigh low levels of fertility.

For the most part, the findings of our field-cage experiments were generally supported by the results of laboratory-cage experiments, although with clear differences in the numbers of eggs produced, which was consistently lower in field cages, and the effects of mating treatment on male competitiveness. The placement and density of ovitraps and the temperature range of ovitrap water clearly differed in field and laboratory cages, which may have affected female oviposition responses and likely reduced egg production in field cages [46]. In terms of male competitiveness, for *Ae. albopictus* mating treatment had a significant effect in field cages but not in the laboratory. This suggests that *Ae. albopictus* males were more attractive to females under semi-field conditions or that females were less likely to discriminate against irradiated males in field cages. However, no such effects were observed in *Ae. aegypti*, although the influence of the irradiation dose was only significant under field-cage conditions in this species, which may also suggest improved perception of male quality by females in the field cages compared to the laboratory setting, as observed in tephritid fruit flies [47]. The female’s ability to assess male quality could be influenced by incident light wavelengths and intensity, changes in the sun’s position, temperature, and humidity, all of which are quite different in field cages compared to a laboratory environment [47]. For example, the fitness costs of genetic modification of a strain of *Ae. aegypti* only became fully apparent in field-cage trials, which contradicted previous findings from laboratory-cage studies [48]. Additionally, cage size alone affected the percentage of egg hatch in *Ae. albopictus* [16] and male competitiveness of *An. arabiensis* in the laboratory [49]. This again underlines the necessity to confirm laboratory findings on male competitiveness with careful field-cage studies under near-natural conditions [11,19,42].

## 5. Conclusions

From the results of our study, we conclude that pilot-scale field tests should proceed with the aim of achieving a 1:10 ratio of fertile:irradiated males or more in treated areas for *Ae. aegypti* and *Ae. albopictus* males previously irradiated with 50 and 40 Gy, respectively. Our results suggest the possibility of reducing irradiation doses for better performance of sterile males. These pilot-scale tests will shortly be undertaken in southern Mexico, for which baseline data on mosquito population dynamics have already been collected [50].

## Figures and Tables

**Figure 1 insects-12-00145-f001:**
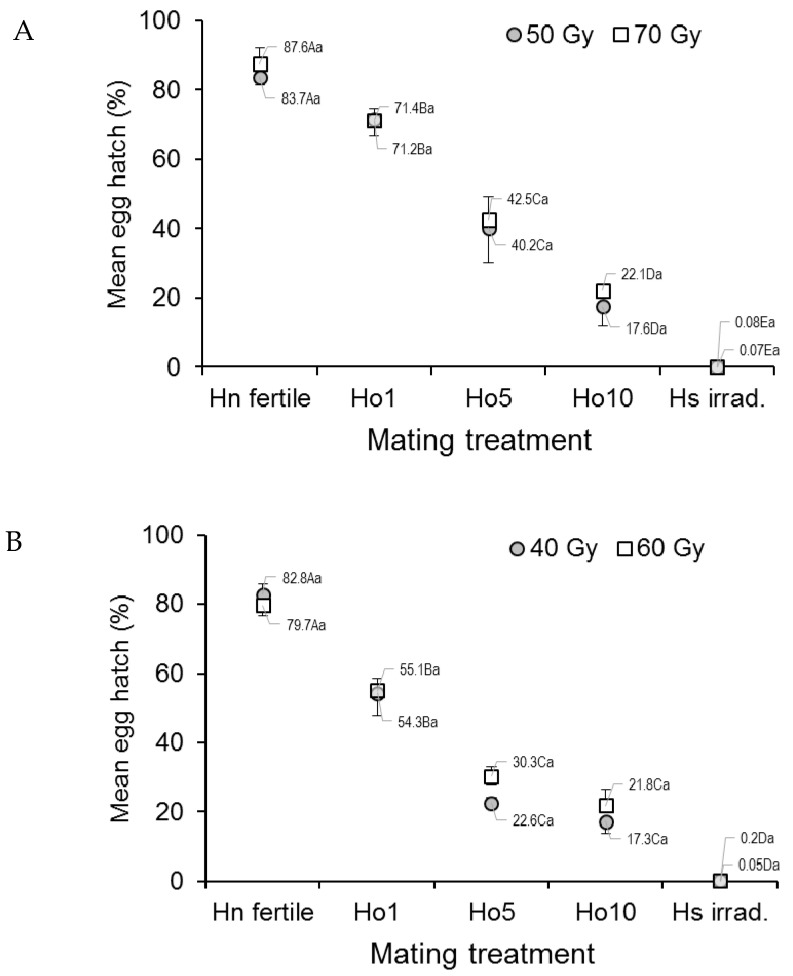
Prevalence (%) of hatching in eggs from (**A**) *Aedes aegypti* and (**B**) *Aedes albopictus* females that mated in laboratory cages under different mating treatments. Males were previously exposed to one of two different irradiation doses in the pupal stage. Controls included only fertile males (Hn) and only irradiated males (Hs). Values next to points indicate mean percentages. Means followed by identical letters did not differ significantly for comparisons among mating treatments of the same dose (upper case) and for comparisons of doses within the same mating treatment (lower case) (ANOVA, Tukey *p* > 0.05). Vertical bars indicate the standard error. For clarity, only half the error bar is shown in some cases.

**Figure 2 insects-12-00145-f002:**
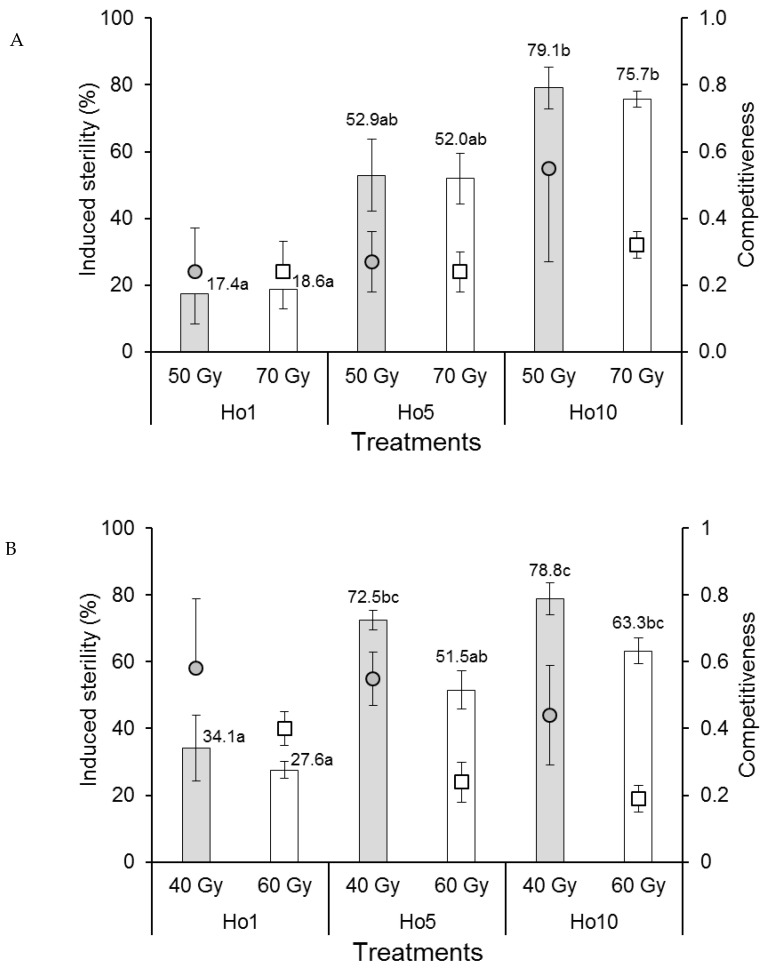
Mean male sexual competitiveness index values (circular and square points) and mean percentage induced sterility of eggs (columns) for (**A**) *Aedes aegypti* and (**B**) *Aedes albopictus* females that mated in laboratory cages with males that had been exposed to one of two doses of radiation in the pupal stage. Controls included only fertile males (Hn) and only irradiated males (Hs). Values above columns indicate mean percentages of egg sterility. Means followed by identical letters did not differ significantly for comparisons among mating treatments and doses (ANOVA, Tukey *p* > 0.05). Vertical bars indicate the standard error. For clarity, only half the error bar is shown in some cases.

**Figure 3 insects-12-00145-f003:**
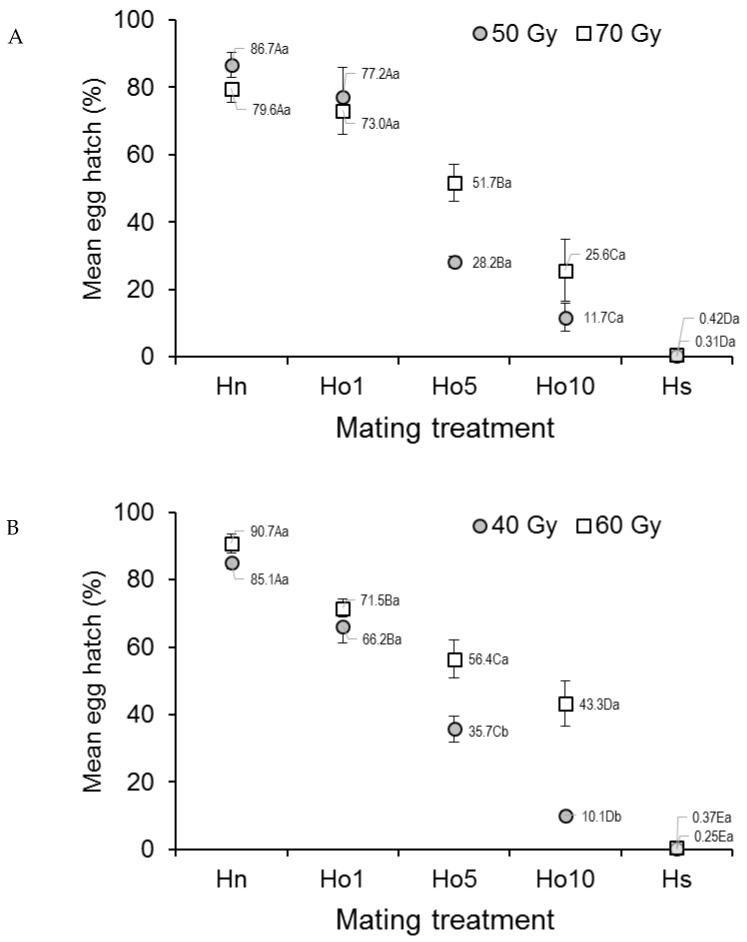
Prevalence (%) of hatching in eggs from (**A**) *Aedes aegypti* and (**B**) *Aedes albopictus* females that mated in field cages under different mating treatments. Males were previously exposed to one of two different irradiation doses in the pupal stage. Controls included only fertile males (Hn) and only irradiated males (Hs). Values next to points indicate mean percentages. Means followed by identical letters did not differ significantly for comparisons among mating treatments of the same dose (upper case) and for comparisons of doses within the same mating treatment (lower case) (ANOVA, Tukey *p* > 0.05). Vertical bars indicate the standard error. For clarity, only half the error bar is shown in some cases.

**Figure 4 insects-12-00145-f004:**
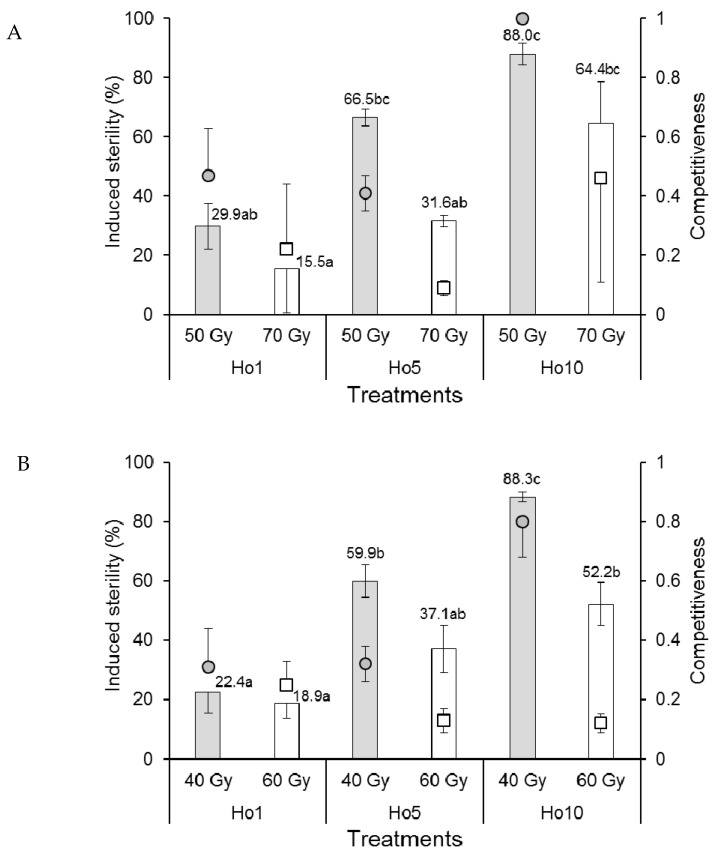
Mean male sexual competitiveness index values (circular and square points) and mean percentage induced sterility of eggs (columns) for (**A**) *Aedes aegypti* and (**B**) *Aedes albopictus* females that mated in field cages with males that had been exposed to one of two doses of radiation in the pupal stage. Controls included only fertile males (Hn) and only irradiated males (Hs). Values above columns indicate mean percentages of egg sterility. Means followed by identical letters did not differ significantly for comparisons among mating treatments and doses (ANOVA, Tukey *p* > 0.05). Vertical bars indicate the standard error. For clarity, only half the error bar is shown in some cases.

**Table 1 insects-12-00145-t001:** Prevalence (%) of insemination of (A) *Aedes aegypti* and (B) *Aedes albopictus* females in laboratory cages subjected to different mating treatments and egg production in mated females. Males were irradiated in the pupal stage with one of two irradiation doses or were fertile controls.

Species, Mating Treatment	Dose (Gy)	Mean Female Insemination ± SE (%) ^1^	Mean Egg Production/female ± SE ^1^
A: *Ae. aegypti*			
Hn (fertile)	−	83 ± 3 a	34.4 ± 3.8 ab
Ho1	50	77 ± 13 ab	33.5 ± 2.9 ab
Ho5	50	80 ± 6 ab	33.4 ± 1.9 ab
Ho10	50	90 ± 6 a	40.4 ± 2.3 a
Hs (irrad.)	50	63 ± 7 b	24.0 ± 3.6 b
Hn (fertile)	-	83 ± 3 a	34.8 ± 0.9 ab
Ho1	70	67 ± 9 ab	34.0 ± 0.9 ab
Ho5	70	63 ± 12 ab	31.0 ± 1.6 ab
Ho10	70	70 ± 12 ab	38.2 ± 3.0 a
Hs (irrad.)	70	50 ± 6 b	31.3 ± 0.8 b
B: *Ae. albopictus*			
Hn (fertile)	−	77 ± 3 a	44.4 ± 2.7 a
Ho1	40	57 ± 3 b	34.3 ± 8.1 ab
Ho5	40	63 ± 9 ab	33.9 ± 9.7 ab
Ho10	40	73 ± 3 a	41.3 ± 5.9 a
Hs (irrad.)	40	50 ± 6 b	23.9 ± 1.6 b
Hn (fertile)	−	73 ± 3 a	34.2 ± 3.9 ab
Ho1	60	53 ± 3 b	26.1 ± 3.5 ab
Ho5	60	60 ± 6 ab	28.5 ± 6.4 ab
Ho10	60	70 ± 6 a	42.8 ± 12.1 a
Hs (irrad.)	60	43 ± 3 b	20.0 ± 0.3 b

Hn (fertile): fertile controls, 1:1; 50 fertile females + 50 fertile males. Ho1: 1:1:1; 50 fertile females + 50 fertile males + 50 irradiated males. Ho5: 1:1:5; 50 fertile females + 50 fertile males + 250 irradiated males. Ho10: 1:1:10; 50 fertile females + 50 fertile males + 500 irradiated males. Hs (irrad.): irradiated male controls, 1:1; 50 fertile females + 50 irradiated males. ^1^ Values followed by identical letters do not differ significantly for comparisons among mating treatments within each dose (ANOVA, Bonferroni test, *p* > 0.05).

**Table 2 insects-12-00145-t002:** Mean percentage of insemination of (A) *Aedes aegypti* and (B) *Aedes albopictus* females in field cages subjected to different mating treatments and egg production in mated females recovered from cages. Males were irradiated in the pupal stage with one of two doses of irradiation or were fertile male controls (Hn).

Species, Mating Treatment	Dose (Gy)	Mean Female Insemination ± SE (%) ^1^	Mean Egg Production/Female ± SE ^1^
A: *Ae. aegypti*			
Hn (fertile)	−	83 ± 3 ab	21.9 ± 1.1 a
Ho1	50	70 ± 6 ab	13.5 ± 3.0 ab
Ho5	50	73 ± 7 ab	18.0 ± 1.5 ab
Ho10	50	87 ± 3 a	22.1 ± 3.4 a
Hs (irrad.)	50	60 ± 6 b	15.2 ± 1.7 b
Hn (fertile)	−	73 ± 12 ab	19.2 ± 0.6 a
Ho1	70	63 ± 3 ab	17.5 ± 0.8 ab
Ho5	70	67 ± 9 ab	16.2 ± 0.4 ab
Ho10	70	77 ± 9 a	16.5 ± 1.0 a
Hs (irrad.)	70	53 ± 12 b	11.6 ± 1.8 b
B: *Ae. albopictus*			
Hn (fertile)	−	60 ± 6 a	30.4 ± 2.5 a
Ho1	40	47 ± 3 ab	22.4 ± 1.3 b
Ho5	40	53 ± 3 ab	22.0 ± 0.9 ab
Ho10	40	67 ± 92 a	24.8 ± 3.9 ab
Hs (irrad.)	40	37 ± 3 b	15.2 ± 4.0 c
Hn (fertile)	−	70 ± 12 a	24.1 ± 0.5 a
Ho1	60	43 ± 9 ab	17.6 ± 0.8 b
Ho5	60	47 ± 12 ab	19.2 ± 0.7 ab
Ho10	60	77 ± 9 a	23.9 ± 0.7 ab
Hs (irrad.)	60	40 ± 6 b	8.5 ± 2.7 c

Hn (fertile): fertile controls, 1:1; 50 fertile females + 50 fertile males. Ho1: 1:1:1; 50 fertile females + 50 fertile males + 50 irradiated males. Ho5: 1:1:5; 50 fertile females + 50 fertile males + 250 irradiated males. Ho10: 1:1:10; 50 fertile females + 50 fertile males + 500 irradiated males. Hs (irrad.): irradiated male controls, 1:1; 50 fertile females + 50 irradiated males. ^1^ Values followed by identical letters do not differ significantly for comparisons among mating treatments within each dose (ANOVA, Bonferroni test, *p* > 0.05).

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
