# Peer review of "Sexual Competitiveness and Induced Egg Sterility by Aedes aegypti and Aedes albopictus Gamma-Irradiated Males: A Laboratory and Field Study in Mexico"

_insects, 2021, doi:10.3390/insects12020145_

Round 1
Reviewer 1 Report
Very interesting paper, clear and well written. I made just two comments in the paper, referred to the size of the cages used in the first competitiveness experiment (Laboratory cages ):
- In the Laboratory cages trial, Authors did not give the size of the cages. I will be important if they will provide size, shape and structure of the cages.
- Just an observation: I am assuming the Authors used the same cage size for all the treatments, even if the mosquito density/cage is very different among the Hn treatments. Comparing the results of the two experiments, I noticed that the laboratory cage data are following a similar trend of the field cage data, conferring that the experiments are compatible and well designed.

Author Response
Response to Reviewer 1 Comments
Point 1. In the Laboratory cages trial, Authors did not give the size of the cages. I will be important if they will provide size, shape and structure of the cages:
We apologize for this omission. The size and characteristics of laboratory cages have now been described: "Cages of 30 x 30 x 30 cm comprised an acrylic frame with nylon mesh walls (BugDorm 1, Taichung, Taiwan)." (lines 135-136)
Point 2A. Just an observation: I am assuming the Authors used the same cage size for all the treatments, even if the mosquito density/cage is very different among the Hn treatments:
Yes, laboratory cage dimensions now stated as identical at two points in Section 2.3.1.
Line 144-145. We add the phrase "The remaining females..." as requested by the Reviewer on the manuscript.
Point 2B. Comparing the results of the two experiments, I noticed that the laboratory cage data are following a similar trend of the field cage data, conferring that the experiments are compatible and well designed:
Thank you for your positive comments. We have also added additional text requested by another Reviewer to address aspects of the differences between laboratory and field-cage findings (Discussion, lines 543-564).

Reviewer 2 Report
The MS about the Sexual competitiveness and induced egg sterility by 2 Aedes species is an interested paper with original data and appears to be technically well executed.
Suggestions to authors:
Introduction
The aim of the study is not clear and need to be rephrased (the last two paragraphs). Please, try to explain the difference between the current study and your previous study (reference No 17). What are the new scientific questions that current study is trying to answer? Additionally, what are the similarities and differences with the reference study No 19.
Results/Discussion
An interesting finding is that there are differences between laboratory and field studies in mean female insemination, mean egg production and competitiveness. Can you discuss these differentiations? Do you think that field conditions affected females’ behaviour? Especially for mean egg production in the field was decreased dramatically. How these results are useful to your future SIT trials?
Author Response
Response to Reviewer 2 Comments
Point 1. Introduction
(1) The aim of the study is not clear and need to be rephrased (the last two paragraphs):
As requested, the final two paragraphs of the Introduction have been reorganized and rewritten in parts for an improved flow of ideas and improved clarity (lines 83-101). An additional reference has been inserted.
(2) Please, try to explain the difference between the current study and your previous study (reference No 17):
Additional details on the extent of the previous study [17] have now been included in the two final paragraphs of the Introduction. The text now reads: "In a previous study, we determined the irradiation dose - fertility response in Mexican colonies of Ae. aegypti and Ae. albopictus when irradiated in the pupal stage [17]. Doses of 50 and 70 Gy resulted in 0.6 and 0% male fertility, respectively, in Ae. aegypti, whereas doses of 40 and 60 Gy resulted in 0.9 and 0% fertility of Ae. albopictus males, respectively. Adult mosquito survival and flight ability were not adversely affected by these doses of irradiation under laboratory conditions in either species [17]." (lines 88-93).
(3) What are the new scientific questions that current study is trying to answer?
We modified the statement on the aims of the current study as follows: "Having established the range of sterilizing doses of irradiation for these species [17], the present study aimed to assess the effects of sterilizing irradiation doses on mating competitiveness and induced egg sterility of Ae. aegypti and Ae. albopictus at differing ratios of fertile males: irradiated males in both laboratory and field cages conditions. The study was performed on the Mexican populations of these mosquitoes as irradiated male mating competitiveness can differ between insect populations, so that the findings from one region may differ quantitatively or qualitatively from those in geographically distant regions [11, NEW REF22]." (lines 94-101).
(4) Additionally, what are the similarities and differences with the reference study No 19:
We have addressed this issue in the Discussion. A new paragraph of text has been inserted specifically to describe the key differences and similarities of our study and the detailed study of Bellini et al. [19] (lines 508-519).
Point 2: Results/Discussion
(1) An interesting finding is that there are differences between laboratory and field studies in mean female insemination, mean egg production and competitiveness. Can you discuss these differentiations? (2) Do you think that field conditions affected females’ behaviour? Especially for mean egg production in the field was decreased dramatically:
We agree. These are two valuable points that we have addressed in detail a new paragraph of text in the Discussion, containing four additional references (lines 543-564).
(3) How these results are useful to your future SIT trials?
The value of the findings is stated concisely in the Conclusions (section 5): "From the results of our study we conclude that pilot-scale field tests should proceed with the aim of achieving a 1:10 ratio of fertile: irradiated males or more in treated areas for Ae. aegypti and Ae. albopictus males previously irradiated with 50 and 40 Gy, respectively. Our results suggest the possibility of reducing irradiation doses for better performance of sterile males." (lines 567-573).
We believe that additional clarification is not required as this provides a clear take-home message from our study, but will be happy to hear the opinion of the Reviewer if this deviates from ours.
